# e-Health Interventions Targeting Pain-Related Psychological Variables in Fibromyalgia: A Systematic Review

**DOI:** 10.3390/healthcare11131845

**Published:** 2023-06-25

**Authors:** Valeria Donisi, Annalisa De Lucia, Ilenia Pasini, Marialuisa Gandolfi, Vittorio Schweiger, Lidia Del Piccolo, Cinzia Perlini

**Affiliations:** 1Section of Clinical Psychology, Department of Neuroscience, Biomedicine and Movement Science, University of Verona, 37134 Verona, Italy; 2UOC Neurorehabilitation, Department of Neurosciences, Biomedicine and Movement Sciences, Neuromotor and Cognitive Rehabilitation Research Centre (CRRNC), University of Verona, Policlinico GB Rossi, 37134 Verona, Italy; 3Pain Therapy Centre, Department of Surgery, Dentistry, Maternal and Infant Sciences, Verona University Hospital, Policlinico GB Rossi, 37134 Verona, Italy

**Keywords:** fibromyalgia, e-Health, m-Health, psychological intervention, pain-related psychological variable, cognitive behavioral therapy

## Abstract

There is growing evidence to support the potential benefit of e-Health interventions targeting psychosocial outcomes and/or pain-related psychological variables for chronic pain conditions, including fibromyalgia syndrome (FMS). This systematic review aims at providing an in-depth description of the available e-Health psychological and/or multicomponent interventions for patients with FMS. Searches were made in PubMed, Cochrane, Web of Science, and PsycINFO up to 15 May 2023, finally including twenty-six articles. The quality of the included articles was medium–high (average quality assessment score of 77.1%). 50% of studies were randomized controlled trials (RCTs) (*n* = 13), and the majority of them focused exclusively on adult patients with FMS (*n* = 23) who were predominantly female. Four categories of e-Health modalities were identified: web-based (*n* = 19), mobile application (m-Health) (*n* = 3), virtual reality (VR) (*n* = 2), and video consulting (*n* = 2). Interventions were mainly based on the cognitive behavioral therapy (CBT) approach (*n* = 14) and mostly involved contact with a healthcare professional through different digital tools. Overall, a growing number of psychological and multicomponent interventions have been created and delivered using digital tools in the context of FMS, showing their potentiality for improving psychosocial outcomes and pain-related psychological variables. However, some digital tools resulted as underrepresented, and the literature on this topic appears highly heterogeneous precluding robust conclusions.

## 1. Introduction

Fibromyalgia syndrome (FMS) is a chronic disease characterized by widespread musculoskeletal pain and associated with other highly disabling symptoms such as fatigue, poor sleep, cognitive dysfunction, depression, and anxiety [1], resulting in a significant reduction in health-related quality of life (HrQoL) [2]. Although still unclear, the causes of this condition involve biological, genetic, and environmental factors as well as psychological ones [3]. Thus, the scientific community supports the need for a biopsychosocial perspective to address the multidimensional needs of people with FMS, adopting an integrated approach that includes pharmacological and non-pharmacological (primarily psychological) treatments [4,5]. Among the psychological interventions for FMS, face-to-face cognitive behavioral therapies (CBT) stand out as the most effective treatment [6]. However, especially in chronic pain, accessibility to psychological treatment programs is often limited by healthcare resources and treatment availability, geographical barriers, and cost [7]. In that sense, the recent development and growth of e-Health, defined as “the use of information and communications technology in support of health and health-related fields” [8], has been recognized as an opportunity to increase efficiency, accessibility, and quality of care [9,10]. The interest in e-Health tools has substantially increased during the COVID-19 pandemic in the different health fields, including investment in technical infrastructure and training for health professionals. In many cases, e-Health tools have become necessary to continue providing health care [9].

The range of e-Health interventions and digital applications is broad and continues to evolve. Different e-Health tools have been introduced and created for chronic pain management. Slattery and collaborators [11] included examples of internet-based interventions, in which patients are engaged by using a broader set of technological devices (i.e., computers or mobile devices, telephone-support, interactive voice response technology, virtual reality (VR), video teleconferencing, and mobile phone applications) [11]. A recent systematic review showed that VR technology produces positive results on a wide range of outcomes, particularly pain relief and functioning [12]. Focusing on internet-delivered psychological treatments (IPTs) for chronic pain, a Cochrane review including different pain conditions highlights that IPTs improved pain, disability, depression, and anxiety in participants with non-headache conditions (both at post-treatment and follow-up). However, among the fifteen randomized controlled trials (RCTs) included, only one study regards FMS, and considerable uncertainty remains about the effect estimates [13]. Similarly, a subsequent review involving twenty-two RCTs of IPTs for chronic pain (two studies regarding FMS) showed small effect sizes for disability, pain intensity, and catastrophizing [14].

To the best of our knowledge, until now, only one systematic review has focused specifically on IPTs for FMS patients [15]. The authors included six RCTs using different types of internet-based cognitive behavioral therapies (ICBTs). Post-treatment clinically relevant benefits in reducing negative mood and disability (regarding functional impairment) were reported compared to a waiting list, treatment as usual, and controls. Based on the best evidence available [6], the review searched for interventions up to January 2018, and the authors restricted the inclusion to trials. Only primarily psychological cognitive behavioral therapies (CBTs) and e-Health modalities where the internet had been utilized to interact with the user were included.

Considering that the body of the research literature in the field of e-Health tools has resulted in higher and broader interest in the past five years, and in particular after the recent pandemic, the current paper updates and enlarges the scope of this previous systematic review, including all the potential e-Health psychological and multicomponent interventions targeting psychosocial outcomes and pain-related psychological variables for patients with FMS. Based on these premises, the aims of the current review are threefold:To provide a systematic and in-depth description of the available e-Health tools delivering multicomponent and psychosocial interventions targeting psychosocial outcomes and/or pain-related psychological variables for patients with FMS;To describe the main psychological approaches used in those e-Health interventions, their structure, and their main characteristics, along with the main psychosocial outcomes and pain-related psychological variables targeted in the included interventions;To describe the impact of the e-Health tools in terms of signals of efficacy, feasibility, and acceptability.

## 2. Materials and Methods

This review was carried out following the “Preferred Reporting Items for Systematic reviews and Meta-Analyses” (PRISMA) guidelines [16].

Three research questions guided the current review:What e-Heath tools are under investigation to deliver psychological and/or multicomponent interventions targeted psychosocial outcomes and/or pain-related psychological variables in patients with FMS?What are the main characteristics of those e-Health interventions in terms of underlying psychological approaches, structure, and addressed outcomes?What is the impact of such e-Health tools in terms of signals of efficacy, feasibility, and acceptability?

### 2.1. Searches

We systematically searched four electronic databases (PubMed, Cochrane, Web of Science, and PsycINFO) up to 15 May 2023. We used the following search strategy: (telemedicine OR tele-medicine OR tele-health OR telehealth OR m-Health OR e-Health OR eHealth OR internet OR web OR online OR mobile OR app OR computer* OR technolog* OR virtual reality OR augmented reality) AND (fibromyalgia OR fibrositis OR FMS (see Appendix A for the full research queries). Two reviewers (A.D.L., I.P.) independently screened record titles and abstracts using the Systematic Reviews Web application Rayyan [17]. They assessed the full texts that were considered potentially eligible by at least one of the two reviewers. A third reviewer (V.D. or C.P.) was involved in the case of dissent.

### 2.2. Inclusion and Exclusion Criteria

Inclusion and exclusion criteria according to the PICOs model are specified in Table 1.

Only articles written in English with no restriction regarding the year of publication were included. When different studies focused on the same intervention were retrieved during the screening procedure (i.e., further analysis of the same e-Health intervention subsequently published by the same research group), we included only the first published study. However, when relevant data were reported in subsequent publications (i.e., “secondary analyses papers”), these papers were excluded, but, if present, the relevant information was considered in the narrative synthesis of the results.

The following types of studies were excluded: systematic review, narrative review, meta-analysis, bibliometric analysis, letter, case-study, book/book chapter, comment, editorial, congress abstract or symposium, poster presentation, and dissertation.

### 2.3. Risk of Bias Assessment

All eligible studies were evaluated against the 16-item quality assessment tool (QATSDD) [18]. The tool demonstrates good validity and reliability for evaluating the quality of sets of research papers adopting various methodologies (e.g., qualitative and quantitative). It consists of 16 criteria, each of which is graded from 0 (meaning “not at all”) to 3 (meaning “complete”). For qualitative or quantitative research, the highest score is 42; for mixed-method studies, the maximum score is 48. For each included article, we reported the score given to each item and the paper’s overall quality score (i.e., resulting from the sum of individual scores for each indicator). Moreover, in addition to the average quality score for all papers, each item’s mean and standard deviation were calculated to describe the items with higher and lower values. Two independent raters (A.D.L., I.P.) assessed the quality of the included studies using the QATSDD. Any potential dissent was discussed, including a third rater (V.D. or C.P.) to adjudicate.

### 2.4. Data Extraction and Synthesis

A systematic and in-depth description of the following data was conducted: study design; characteristics of the sampled population for age, gender, and presence of psychiatric diagnosis among the criteria; the intervention outcomes and pain-related psychological variables; the follow-up duration of the study, when applicable; when involved, the type of control group (reported in Appendix B); the type of e-Health tools used for delivering (even partially) the intervention; and the type of intervention, its conceptual basis, structure, main components, duration, and format). Moreover, signals of efficacy, feasibility, and acceptability were summarized considering the available results (even in secondary analyses papers). It should be specified that this article aims at giving a broad overview of the current state of knowledge on signals of efficacy in this specific field. For signals of efficacy, we considered results on the impact of the interventions on different outcome measures without claiming to provide a definitive or peremptory conclusion in that regard. As for feasibility and acceptability, any information reported on those aspects was synthetized.

Two reviewers (A.D.L., I.P.) extracted data from the selected studies using a data-collection form in Microsoft Excel. Doubts were discussed, and any disagreement about study eligibility was resolved by a third reviewer (V.D. or C.P.).

## 3. Results

### 3.1. Study Selection

The electronic literature search yielded 2698 records in total, with 777 duplicates that were removed. During the study selection process, 1921 records were analyzed by title and abstract, and 1831 were excluded according to the inclusion and exclusion criteria. Finally, 92 records were selected for the full-text analysis, of which 66 were excluded for various reasons (see flow-chart—Figure 1).

Among the excluded articles, nine articles were excluded because they focused on e-Health interventions already described in a previous article by the same research group. In these cases, the first paper presenting the intervention was included in the review (i.e., defined “primary paper”), while the subsequent papers were excluded from the flowchart (Figure 1) and whenever we reported quantitative information on the number of papers. However, when relevant data were reported in these so called “secondary analyses papers”, they were inserted in Table A1 [19,20,21,22,23,24,25,26,27] and in the corresponding sections of the results.

Twenty-six studies finally met the inclusion criteria [28,29,30,31,32,33,34,35,36,37,38,39,40,41,42,43,44,45,46,47,48,49,50,51,52,53].

### 3.2. Characteristics of the Included Studies

Appendix B shows the main characteristics of the included studies. The studies were published between 2008 and 2022 and were mainly conducted in Europe (Norway [31]; Spain [32,37,40,41,48,49,50,51]; Switzerland [33]; Sweden [36,44]; Italy [53]) and in the USA [28,29,34,35,38,45,46,52]. Three studies were conducted in Canada [39,43,47], one in South Africa [30], and one in Brazil [42].

50% of the included studies (*n* = 13) were randomized controlled trials (RCTs) [28,29,31,34,35,37,39,41,43,44,47,49,50]. Three studies were pre–post-test studies [32,36,51]; four were observational studies [33,38,46,53]; three were feasibility studies [40,42,45]; and three studies were research protocols [30,48,52]. Only three of those studies also reported qualitative data [40,42,48].

### 3.3. Selected Populations in the Included Studies

Twenty-three studies focused exclusively on patients with FMS [29,30,32,33,34,35,36,37,38,39,41,42,43,44,45,46,47,48,49,50,51,52,53]. Three studies included mixed-population groups, but at least 50% of FMS patients were included [28,31,40].

Regarding the criteria used for the diagnosis of FMS, in 12 studies [29,32,35,37,38,41,43,45,49,50,51,53], FMS was diagnosed according to the American College of Rheumatology (ACR) classification criteria [1,54,55]. In five studies, the FMS diagnosis was required to be established by a health professional [28,33,40,42,46]. In the study by Kristjánsdóttir et al. [31], both previous criteria were used for the inclusion of patients; however, 83% of participants met the ACR classification criteria for FMS. The remaining four studies relied on self-reported FMS diagnosis [34,39,44,47].

When reported, the percentage of female patients in the studies ranged between 68% and 100%, and the age range of the population varied between 18 and 89, except for the study by de la Vega et al. [40], which included younger people aged 13–24 years.

Sixteen studies excluded patients with severe psychiatric disorders [29,30,31,32,34,35,36,37,38,41,43,44,45,47,51,53] plus the study by Yuan & Marques [42], according to the secondary analyses study of the authors [24], and one study excluded participants with moderate or severe cognitive impairment [52]. Seven studies included patients with different mental health conditions [29,32,35,39,44,47,49]. The remaining studies did not mention mental health or psychological symptoms within the exclusion criteria to define the sample.

### 3.4. Risk of Bias Assessment

Overall, the %QATSDD score ranged between 47.6% (mean raw score = 20) [42] and 92.9% (mean raw score = 39) [44]. The average quality score for all papers was 77.1% (raw score of 32.4). Only one study resulted in under 50% of the total score [42]. Variations in quality among the studies mainly concerned the following items: evidence of consideration of sample size in terms of analysis; the presence of a representative sample of a target group of reasonable size; the presence of a statistical assessment of the reliability and validity of measurement tool(s); and a critical discussion of the strengths and limitations. The lowest QATSDD single-item score referred to user involvement in the design of the study, with only six studies reporting it (item mean score ± SD = 0.48 ± 1.00) [31,33,40,42,43,44]. In addition, the explanation of the rationale for the choice of data-collection tools was very limited in most studies (item mean score ± SD = 1.8 ± 0.83) (see Table 2).

### 3.5. e-Health Interventions for Patients with FMS

#### 3.5.1. e-Health Modalities

Regarding the type of e-Health modalities used for the interventions, four main categories were identified: web-based, m-Health, virtual reality (VR), and video consulting (see Figure 2). In 19 studies, the treatment program was delivered through a web-based modality (i.e., delivered via websites or web applications) [28,29,33,34,35,36,37,38,39,41,43,44,45,46,47,48,49,50,52]. In some cases, participants were required to access a website to complete several learning modules aimed at providing information about FMS and teaching self-management strategies (e.g., [28,29,37,39,48]) or skills targeting specific psychological aspects, such as positive affect [41,52] and psychological flexibility, symptom-related fear, and consequent avoidance behaviors (e.g., [36,43,44]) or symptoms of depression and anxiety [35]. Moreover, one study delivered a computer-based program via DVD format or USB flash drives since it was not available via the internet during the study implementation [38]. In the study by Davis & Zautra [34], participants received an email with a link to each module, and the material in the modules was delivered via eLearning software; similarly, in the study by Serrat et al. [49], participants received an email with a link to a video hosted on a private YouTube channel. In the study by Friedberg et al. [45], participants were land-mailed to a bilateral stimulation and desensitization (BSD) intervention (i.e., video, written instructions, and an mp3 file of the audio BSD technique) with login-based online diaries to track pain, fatigue, and intervention use.

In Carleton et al. [47], as part of an attention bias modification (ABM) program, the authors utilized a specific web software to administer the attention tasks.

Two studies proposed mobile phone applications: ProFibro—a mobile multicomponent app for the promotion of self-care and improvement of symptoms and HrQoL in patients with FMS involving several functions, such as patient education through animation, self-monitoring, and sleep strategies [42]; and Fibroline—a mobile app with a self-administered CBT program for juvenile fibromyalgia syndrome (JFS) or chronic widespread pain (CWP) [40]. Similarly, Kristjánsdóttir et al. [31] used mobile software to deliver an ACT-based intervention, diary completion, and daily written feedback from a therapist.

Two studies applied VR approaches in the context of in-person treatments. Morris et al. [30] investigated VR exposure therapy as a treatment for pain catastrophizing, while Botella et al. [32] investigated a VR program as an adjunct to face-to-face CBT for the delivery of relaxation and mindfulness strategies. In detail, while participants were immersed in the VR environment, the system provided instructions on observing the different elements offered by the scenarios, remaining focused on the present moment and participating in the experience without making any judgments.

Two studies used a video-consulting modality, providing online group psychological- [51] or mind–body-based [53] sessions through different video-meeting platforms.

Overall, looking at the included papers, fifteen studies included some kind of contact with a healthcare professional (e.g., psychotherapists, nurse specialists, etc., [28,31,33,36,37,39,41,43,44,45,49,50,51,52,53]), while in eight studies the interventions were unguided, as they did not involve any type of contact with healthcare professionals, and the participants carried out the intervention’s activities in complete autonomy [29,34,35,38,40,42,46,48]. Finally, three studies foresaw the use of e-Health tools in the context of face-to-face interventions and therefore involved the presence of a therapist [30,32,47].

More specifically, when the web-based or m-Health studies included a kind of interaction, treatment progress was strictly monitored by a therapist or other healthcare professionals with support and guidance functions, with whom participants had direct contact through asynchronous text messages [31,44], SMS, telephone calls, emails [36,43,49,52], or internal messaging systems into the website used for delivering the intervention [33,50]. In Camerini et al. [33], the internal messaging systems also allowed them to communicate with each other on the website. In one case, an automatic interactive system was proposed for the participants to receive automatic individualized feedback from the system based on the data they entered into an online health diary over time [46]. In addition, in Molinari et al. [41], participants received two automatic weekly SMS reminders to practice exercises and reinforcements. In three studies, both types of interactions were automated or delivered by a person [28,37,39].

#### 3.5.2. Main Psychological Approaches and Strategies at the Basis of the Interventions

As reported in Table 3, most of the included studies evaluated interventions based on CBT approach. More specifically, five studies applied CBT-based interventions [29,32,37,39,40], which were usually structured into several modules that broadly included (i) psychoeducation about chronic pain and FMS, (ii) behavioral and cognitive skills designed to help with symptom management, and (iii) prevention of relapse (e.g., [29,37,39,40]). In addition to those contents, some studies provided specific activities to induce positive emotions (e.g., relaxation exercises) and to promote patients’ motivation, self-efficacy, and behavior activation (e.g., [20,21,32]). Within the CBT approach, two interventions were based on exposure therapy, a specific approach generally involving exposing patients to feared stimuli to reduce the person’s fearful reaction and the related avoidance behavior [30,44]. Specifically, Morris et al. [22,30] applied visual exposure to healthy exercise activities to treat exercise-related pain catastrophizing. Similarly, Hedman-Largelof et al. [44] used exposure-based CBT to pain-eliciting situations to break the vicious cycle of preoccupation with symptoms, avoidance, and increased pain.

Three studies based the intervention on the acceptance and commitment therapy (ACT) approach—the third wave of CBT [36,43,51]. The ACT-based interventions emphasized contents related to acceptance, psychological flexibility, and value-based action instead of pain control through mindfulness and behavior-change strategies (e.g., [31,36,43,51]). Similarly, the intervention was based on a mindfulness-based approach [34]. Three studies combined different psychological approaches: ACT and CBT [31]; CBT and interpersonal therapy [35]; and mindfulness and ACT [52].

Molinari and collaborators [41] based the intervention on a positive psychology approach (i.e., positive future-thinking) to augment positive affect and promote positive functioning.

Two studies utilized specific psychological techniques: the bilateral stimulation and desensitization (BSD) technique, focused on pain and fatigue reduction [45]; and the attention bias modification (ABM) program to reduce patients’ hypervigilance for pain-related cues, which, paradoxically, may maintain pain [47].

In six studies, a specific psychological theory/approach was not explicated in the paper. In those cases, the intervention was based on self-management strategies [28,38,46,48], online psychoeducation and informative interventions [33], or the possibility of communicating with the medical or nursing staff at any time through a messaging system, which was also used for providing patients with FMS with properly documented information relating to their disorder and their regular medication schemes [50]. Three studies used a multicomponent intervention based on the above approaches (e.g., psychoeducation, CBT strategies, and physical activity [42,49,53]).

Overall, most interventions (18 studies) provided patients with educational lectures to increase their knowledge about FMS and teach them effective skills designed to facilitate symptoms management combined with homework exercises and diaries to monitor several pain-related aspects.

The intervention time frame ranged from one week (i.e., the proposed app was tested in terms of its quality of use with ten patients) [42] to one year (i.e., patients were given access to an online platform) [50]. The interventions were delivered individually, except for three studies adopting a group format [32,51,53].

#### 3.5.3. Psychosocial Outcomes and Pain-Related Psychological Variables Targeted by the e-Health Interventions

The bar chart below (Figure 3) describes the variables considered in the included papers, which are classified into three main categories: psychological, physical, and integrated (including integrated perceived bio-psycho-social outcomes) variables. Among the psychological ones, we explored pain-related psychological and emotional distress variables. The investigation of the physical outcomes goes beyond the scope of this review, and we simply listed them in Table A1. Among the integrated outcomes, perceived state of health and health-related quality of life; functional impairment and disability, the global impact of FMS on functioning and FMS symptom level, impact, and severity of fatigue on functioning, subjective pain experience, self-care agency; perceived sleep quality and psychological aspects related to sleep were included.

Among the psychosocial outcomes, a large proportion of the studies (*n* = 14) assessed emotional distress such as anxiety [29,31,36,37,39,44,47,49,51], depressive symptoms [29,31,32,36,37,39,41,43,44,47,49,51,52], and positive and negative affect [32,34,41,52]. Twelve studies evaluated pain-related psychological variables such as kinesiophobia, catastrophizing, and coping strategies [30,31,32,34,37,39,41,43,45,48,49,53]. Five studies assessed perceived self-efficacy [28,37,39,41,48]; five studies assessed acceptance/psychological flexibility [31,36,43,44,51]; two studies evaluated mindfulness attitude [43,44]; one study assessed resilience [53]; and one study assessed anxiety sensitivity and illness/injury sensitivity [47]. The outcome measures (i.e., questionnaires) used in the studies are listed in Table A1.

For each e-Health modality, the number of studies targeting the specific variable was reported. In the same study, different variables could be analyzed. 

#### 3.5.4. Signals of Efficacy of the e-Health Interventions

Among the included papers, 13 RCTs compared the e-Health interventions with other control conditions [28,29,31,34,35,37,39,41,43,44,47,49,50]. Looking at the web-based category, Simister et al. [43] found significant improvements in favor of online ACT + treatment as usual (TAU) on measures of FMS impact, depression, and kinesiophobia compared with TAU at post-treatment and three-month follow-up. In Vallejo et al. [37], both the CBT and the web-based CBT (iCBT) interventions showed improvements at post-treatment in psychological distress, depression, catastrophizing, and use of relaxation—unlike the waiting list group (WL)—but only for iCBT self-efficacy, catastrophizing, and helplessness improved at follow-up. The intervention carried out by Davis & Zautra [34] revealed more significant improvements in social functioning, positive affect, and coping efficacy for pain and stress in the online mindful socioemotional regulation group versus the healthy tips control group across the 6-week trial. Friesen et al. [39] showed significant improvements for the iCBT intervention versus the waiting list group on depression and fear of pain at post-treatment and four-week follow-up, with smaller effects on measures of generalized anxiety. Hedman-Lagerlöf et al. [44] reported more significant effects after the web-based exposure therapy in different outcomes (i.e., FMS symptoms and impact, fatigue, general disability, quality of life, depressive symptoms, general anxiety symptoms, insomnia, pain-related distress, non-reactivity to inner experiences, and pain-related avoidance patterns) than the effects obtained in the waiting list group. The differences were also significant at the 12-month follow-up. Similarly, Molinari et al. [41] observed that the web-based positive psychology intervention obtained significant improvements compared with the daily activities control condition on measures of depression, positive affect, and self-efficacy at post-treatment and even also on optimism and negative affect at three-month follow-up. Menga et al. [35] found lower scores on measures of FMS impact in the iCBT program group at 6- and 12-week follow-up. In the study by Garcìa-Perea et al. [50], the web platform group showed constant improvement over the 12-month study on measures of the perceived general state of health, anxiety, and depression relative to the TAU control group. Serrat et al. [49], who delivered the treatment via a video format, observed significant improvements with small-to-moderate effect sizes in the intervention group compared to the TAU control group regarding functional impairment, depression, and anxiety symptoms.

Conversely, in the study of Carleton et al. [47], the web-based ABM program was associated with a small significant improvement in pain experience post-treatment, but not at follow-up and without differences with the control group, which underwent tasks similar to the ABM procedure. Moreover, in Lorig et al. [28], participants reported benefits from the web-based self-management program in health status and self-efficacy measures at one-year follow-up. However, the results were lower than in other diagnostic groups.

Regarding the studies that used research designs other than RCT, signals of efficacy of the e-Health interventions emerged. Among the web-based category, a website providing information about FMS and social support produced positive effects on patients’ health knowledge, which improved self-management and, consequently, reduced the FMS impact, leading to better health outcomes [33]. Collinge et al. [46] observed a significant inverse association between the functional impact of FMS and the use of a self-management program delivered through a website (including online health diaries with automated feedback). Ljótsson et al. [36] found moderate-to-large within-group effect sizes on different outcomes (i.e., measures of FMS symptoms and impact, disability, quality of life, depression, anxiety, fatigue, and psychological flexibility) immediately after the website ACT intervention and at 6-month follow-up. In addition, the computer-based self-management program proposed by Sparks et al. [38] showed a significant reduction in overall FMS impact post-treatment. In the study by de la Coba et al. [51], the ACT online group sessions were associated with a significant reduction in distress, and biopsychosocial impact of FMS and significant improvements in satisfaction with action and emotional discomfort, both post-treatment and at 6-month follow-up. Similarly, in the study by Paolucci et al. [53] the mind–body online group program was associated with a significant reduction in physical and mental distress, fear of movement, and disability both at post-treatment and at 1-month follow-up. The study by Friedberg et al. [45] demonstrated the preliminary feasibility of the BSD intervention administered through the support of video and audio files in reducing pain catastrophizing.

In the pilot study by Botella et al. [32], VR —as an adjunct to face-to-face CBT—produced long-term benefits on measures of depression, positive affect, and healthy coping strategies. The mobile app proposed by de la Vega et al. [40] found significative improvements in measures of pain severity, anxiety, and depressive symptoms in a sample of patients with FMS at post-treatment and three-month follow-up, thus showing the preliminary effectiveness of the results [25]. The mobile software proposed by Kristjánsdóttir et al. [31] resulted in a reduction in pain-related catastrophizing immediately after the treatment period in comparison to the control group, which at 5-month follow-up remained moderate [19].

#### 3.5.5. Feasibility and Acceptability of the e-Health Interventions

In ten studies, feasibility, acceptability, or participants’ satisfaction with the intervention were measured. Among these, in the three feasibility studies, intervention feasibility and usability were assessed based on participants’ feedback obtained through telephone calls [45], individual unstructured interviews after using a prototype mobile app for a week [42], and online surveys after following a mobile phone intervention [45] with positive results.

In the study of Kristjánsdóttir et al. [31], the feasibility of the smartphone intervention was assessed with single questions post-intervention, showing that most of those who completed it found the participation helpful. In the pilot study by Botella et al. [32], participants rated their satisfaction using the VR component in a CBT intervention, showing that this procedure was well-accepted by patients who reported high levels of satisfaction with its use. Similarly, the web-based intervention proposed by Williams et al. [29] resulted in a higher general satisfaction level than the control group. Likewise, in the research conducted by Friesen et al. [39], participants’ satisfaction with the program was high, with 86% reporting being either very satisfied or satisfied with the treatment. However, they expressed the need for assistance in overcoming barriers to completing the program.

Sparks et al. [38] assessed patient perspective on using a web-based intervention with a survey consisting of six multiple-choice and open-ended questions. Although many participants evaluated the intervention as helpful in their self-management, many still preferred receiving health information directly from their healthcare providers.

Finally, two study protocols included a feasibility and acceptability assessment [30,52]. In the first study, specific information was collected using a datasheet; in the second study, feasibility was assessed based on the frequency and descriptive statistics for enrollment rates, number of sessions completed, number of weeks required to complete the intervention, and acceptability through participant feedback.

## 4. Discussion

The present paper systematically reviewed the e-Health tools applied in the context of multicomponent and psychosocial interventions targeting psychosocial outcomes and pain-related psychological variables in patients with FMS. We also reviewed the main psychological approaches at the basis of the interventions and results regarding their efficacy, feasibility, and acceptability. Among the included studies, a wide variety of countries were represented, with the first study having been published around fifteen years ago, confirming the recent interest in this field in different geographic contexts.

Regarding the quality, generally, the studies’ quality was medium–high. The main strengths of the included papers were the detailed description of the data collection procedure, the fit between the stated research question and the method of data collection and analysis selected, and the good justification for the analytical method chosen. The main limits of the studies were the lack of user involvement in the study/intervention design and a very limited explanation for the choice of data-collection tools. These points need to be strengthened in future studies. Patient engagement in research design and intervention development has been recognized as essential to enhancing the adherence to, efficacy, and efficiency of healthcare services provision [56]. Indeed, the participatory approach has been largely used in the field of chronic diseases [57] and e-Health [58,59]. Moreover, the e-Health approach itself presents the potential for engaging patients in their care process [56].

Most of the included studies were specifically designed for individuals with FMS. However, three issues need to be clarified within this disease-specific perspective. First, the patients with FMS examined in the current review were mainly female, according to the large proportion of the diagnosis among the female population [60], and covered different age populations, predominantly in the age range of approximately 40 to 65 years. Although consistent with the relevant scientific literature, these factors make it challenging to generalize the results to the entire FMS population, particularly males and younger or older patients. However, it has to be considered that, regarding the gender disparity within this condition, there is a lack of empirical evidence about the psychological and clinical manifestations of FMS and the available intervention options in the male population. As pointed out by a recent literature review [61], FMS is characterized differently in the two genders, both from a biological and psychological point of view (e.g., FMS males seem to endure pain for longer periods before seeking medical attention, which probably also has socio-cultural factors). Therefore, research on the male population should be further developed, potentially adopting a differentiated approach in the pharmacological and psychological treatment of FMS.

The literature has paid little attention to younger patients. Epidemiological studies indicate that FMS is more common in middle age [62], but it can affect all ages, including adolescents. For example, one study considered a young population aged between 15 and 24, dedicating a CBT mobile app to a small sample of patients with JFS [40].

Finally, plenty of evidence in the medical literature confirmed the high prevalence of several psychiatric comorbidities in FMS (e.g., [63]). However, there was a lack of consistency among the studies in including/excluding patients with psychiatric disease. More than half of the studies appeared to exclude patients with severe psychiatric disturbances, while several studies did not mention this criterion, suggesting that patients with psychiatric diseases may have been included. Seven studies [29,32,35,39,44,47,49] included patients with mild to moderate psychiatric symptoms (e.g., anxiety or depressive symptoms). Although a description of the reason for excluding these patients was not reported in the included studies, the inclusion/exclusion can be guided by stepped care principles according to the type of intervention and the level of need.

Overall, the proposed interventions showed signals of efficacy in the considered psychosocial outcomes. FMS negatively affects physical, psychological, and social functioning, significantly reducing the perceived quality of life. This decrease is related to several factors, such as anxiety, depression, and pain (e.g., [64]). The majority of the included studies focused on different nuances of emotional distress, such as depressive symptoms, anxiety, positive and negative affect, of HrQoL, and perceived functioning. Among the pain-related psychological variables considered to play a central role in maintaining and exacerbating FMS symptoms, pain catastrophizing, fear-avoidance behaviors, self-efficacy, and coping strategies were mostly evaluated. Among those, pain catastrophizing, defined as “an exaggerated negative mental set brought to bear during actual or anticipated painful experience” [65], is a widely observed maladaptive coping strategy in patients with FMS (e.g., [66]). Evidence in the literature showed that higher levels of catastrophizing in FMS are associated with a greater frequency of emotional disturbances (such as anxiety and depression) and more severe forms of pain (e.g., [67]). Regarding perceived self-efficacy, there is evidence that this psychological aspect is crucial in affecting psychological and physical functioning in patients with chronic pain (e.g., [68]). Since patients with FMS tend to have low self-esteem and perceived self-efficacy, interventions to improve those aspects are also needed to promote a better adaptation to the syndrome (e.g., [63]). To sum up, the considered psychological aspects targeted by e-Health tools went beyond the negative mood outcome analyzed by Bernardy et al. [15]. Although the limits of the studies and the differences among the variables/questionnaires considered hamper drawing definitive results and allowing a comparison among the studies, the proposed interventions showed a signal of efficacy on the considered psychosocial outcomes.

In the current review, the included e-Health tools ranged from interventions in which the e-Health approach was only a component of a more extensive integrated intervention (e.g., the use of VR as part of a face-to-face intervention) to others in which the intervention was based only on the e-Health approach (e.g., a web-based program through a website dedicated to self-management or an intervention delivered through video consulting). Specifically, four main categories of e-Health modalities were represented in the current review: web-based intervention (73%)—defined as “self-guided or therapist-assisted programs to improve knowledge, providing support, care, or treatment to a diverse population with a range of health problems” [69]; m-Health (mobile health) (11.5%)—mobile-based or mobile-enhanced programs; video consulting (7.7%)—the use of high-quality real-time video and audio connection via online internet networks; and virtual reality (7.7%)—“a 3-dimensional computer-generated environment that the individual can explore, interact with, and manipulate” [11].

The use of these tools varied depending on the diverse purpose of the intervention. Web-based programs represent useful tools for helping patients to increase their knowledge and self-management about FMS by providing specific learning modules. The tools often included additional homework (such as executing experiential exercises, practicing different skills, keeping health diaries, etc.) to consolidate the new learning. In that regard, web-based tools for chronic pain self-management are well documented (e.g., [70]). At the same time, the accessibility of appropriate and reliable information for patients with FMS is considered useful in this field [50]. In some studies, the web-based interventions allowed an easy interaction with patients with FMS, namely interaction with professionals through telephone calls, text messages, e-mails, and chat or automated program messages/feedback. Although the studies’ heterogeneity in the current review precludes any comparison between these modalities, other reviews suggest that guided internet-delivered interventions might be superior to unguided ones in terms of effectiveness (e.g., [15,71]).

More recently, mobile applications have been developed that show promising effects in persons with chronic pain (e.g., [72]). They allow for an interactive and engaging experience. However, as reported in the current review, a few m-Health solutions have been developed specifically for persons with FMS: ProFibro [42] and Fibroline [40], a mobile software proposing web-based diaries, online exercises, and feedback. This latter solution showed promising results in reducing catastrophizing [31], while ProFibro’s preliminary results did not show significant differences in HrQoL, symptoms, or self-care agency [24]. Both tools were developed in collaboration with FMS end-users, showing high feasibility and acceptability.

Still, preliminary evidence has been reported for video consulting, with the experience of a brief ACT intervention [51] and of a mind–body intervention [53] through video consulting, both reporting improvements in quality of life.

Regarding VR, an adaptive environment with specific content for relaxation and mindfulness skills has been created, with its psychological benefits having been demonstrated more than ten years ago [32]. Two subsequent studies [20,21] confirmed the potential of this VR-based approach as an adjunct to the psychological treatment of patients with FMS with positive benefits on different psychosocial outcomes. In recent years, VR has been strategically used to lessen pain catastrophizing while overcoming any potential limitations of in vivo exposure therapy or simulated exposure therapy. Morris et al. [30] proposed this use of VR for FMS with preliminary positive results [22].

Generally speaking, using e-Health solutions confers several advantages to the administered interventions. Digital intervention programs may be promising alternatives to traditional face-to-face treatments as they allow them to overcome specific logistic barriers (e.g., traveling long distances to reach health-appropriate healthcare networks, especially for people living in rural areas) or health-related barriers (e.g., physical impairments). It increases patients’ accessibility to effective treatments with potentially positive effects on the quality of life and economic costs (e.g., [49,50]). Moreover, the fact that internet-delivered interventions are carried out by the patients themselves in their home environment could facilitate the generalization of treatment gains across contexts and activities. Furthermore, this modality does not imply scheduled appointments; thus, it is heavily flexible for patients and therapists (e.g., [44]). All these aspects make e-Health tools highly cost-effective (e.g., [44,49,50]). However, a cost evaluation in the papers included in the current review was only generally considered in a few papers [32,36,44]. Beyond the effects, when assessing the potentiality of e-Health tools, an important aspect to consider is their feasibility in clinical practice. Only a few of the included studies evaluated feasibility parameters, for example, collecting participants’ feedback on using a prototype mobile app. However, some consideration of the feasibility was reported in the study discussion and represents a helpful suggestion for further research directions in this field. For example, the use of app- and web-based tools requires users to be familiar with technology and access to the internet (e.g., [28,35,50]), or at least it takes time to learn the skills needed to handle these new technologies (e.g., [38]). Additionally, it relies on patients’ ability to plan their activity during the treatment. It can be challenging for therapists to address patients’ concerns about the intervention only through email or written messages (e.g., [44]). More in general, several challenges still exist in terms of hampering its broad implementation into health services (among them are technical issues, e.g., a lack of internet access, privacy, data security, difficulties in performing remote physical examinations or diagnostics, and a lack of training of healthcare professionals [73]).

To conclude, some issues remain open and thus need to be addressed. In particular, further studies should fill the knowledge gap about the clinical characteristics and the most effective treatment options for FMS when it occurs in specific populations, including males, young and old individuals, and patients with psychiatric comorbidities. As for the latter aspect, there is a lack of consistency between the included studies regarding the inclusion/exclusion of patients with psychiatric diseases, generally without providing any reason for this choice. This issue should therefore be addressed in future investigations. Moreover, e-Health potential solutions other than web-based modalities seem underdeveloped in this field, suggesting the need for future research into them. Finally, feasibility parameters and cost evaluations of such tools should be taken into account by future studies.

## 5. Strengths and Limitations

This systematic review presents both strengths and limitations. Regarding its strengths, this systematic review enlarges and updates the results of a previous review on the topic, allowing for a deeper analysis and understanding of the potentiality of e-Health tools for psychosocial benefits for patients with FMS. The rigorous methods used for the search strategy, data analysis, and studies appraisal were based on internationally recognized tools.

Regarding its limitations, the high heterogeneity of the included studies in most of the targeted variables of the review deeply limited our possibility to draw conclusive results. Moreover, the classification and description of the interventions were based on the information reported in the studies, which was often limited. As suggested by Rohn et al. [74], a more accurate description of the contents, platforms, and infrastructure related to these interventions should be included in the e-Health intervention field.

## 6. Conclusions

The present review presents a complete description of the state of the art on use of e-Health strategies targeting psychosocial outcomes and pain-related psychological variables for patients with FMS.

A growing number of psychological and multicomponent interventions using e-Health tools in the context of FSM have emerged, with a significant prevalence of interventions based on a web-based modality. Still, few experiences of m-Health, VR, and video consulting have been implemented in the FMS context. Looking at the extensive range of psychosocial variables targeted by the e-Health tools and the signals of the efficacy of the included interventions, e-Health tools have shown the potential to positively influence psychosocial variables representing the core dimensions of FMS.

We also showed that the existing literature is highly heterogeneous regarding study design, psychosocial outcomes, instruments to assess psychosocial dimensions, and e-Health tools reported across studies, making it difficult to provide robust conclusions. Far from being merely a limitation, such an observation can stimulate a discussion on the need to develop further research in FMS and e-Health intervention. Therefore, future studies should accurately focus on selecting variables and instruments to improve the comparison among results.

## Figures and Tables

**Figure 1 healthcare-11-01845-f001:**
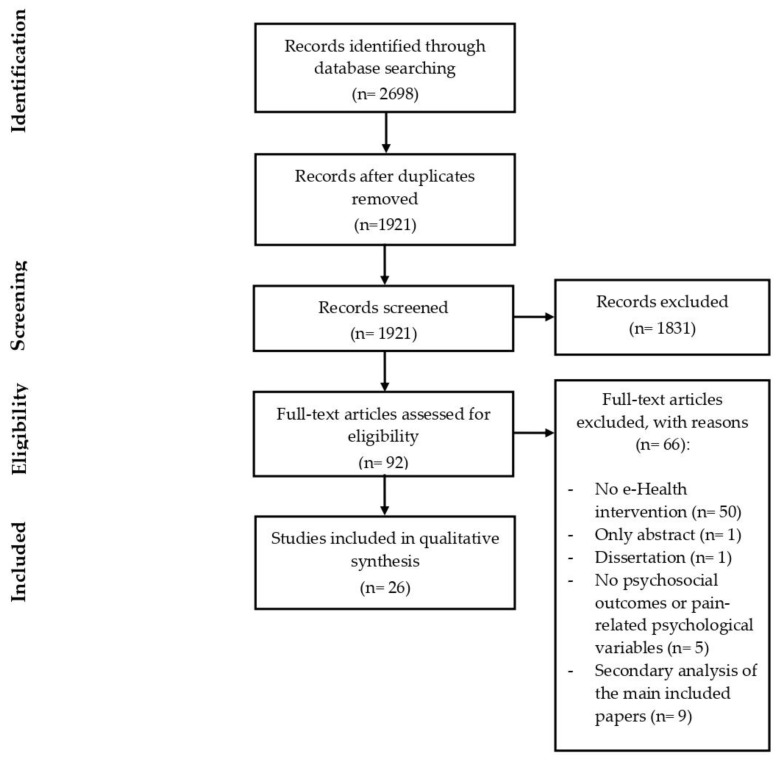
Process of study selection (based on PRISMA flow diagram).

**Figure 2 healthcare-11-01845-f002:**
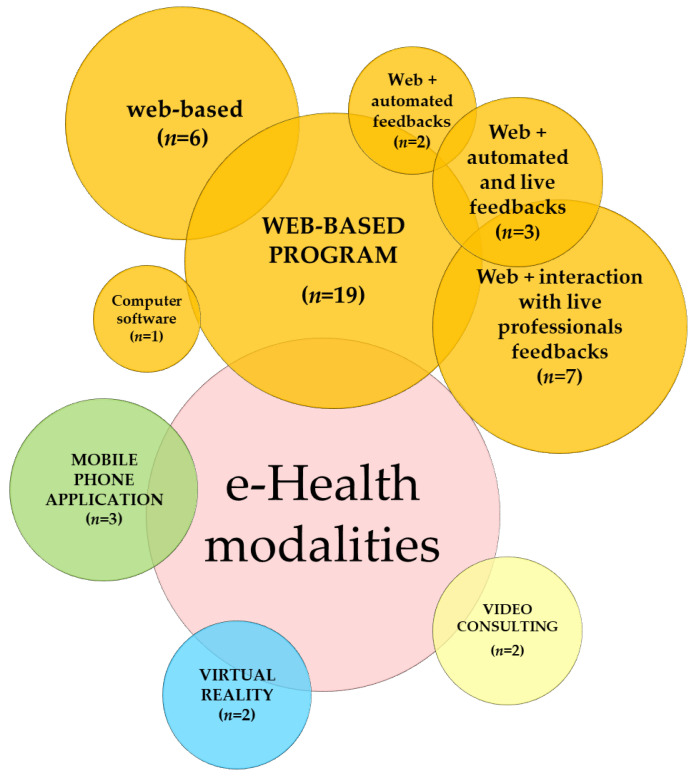
e-Health modalities used to deliver the interventions (*n* = number of studies).

**Figure 3 healthcare-11-01845-f003:**
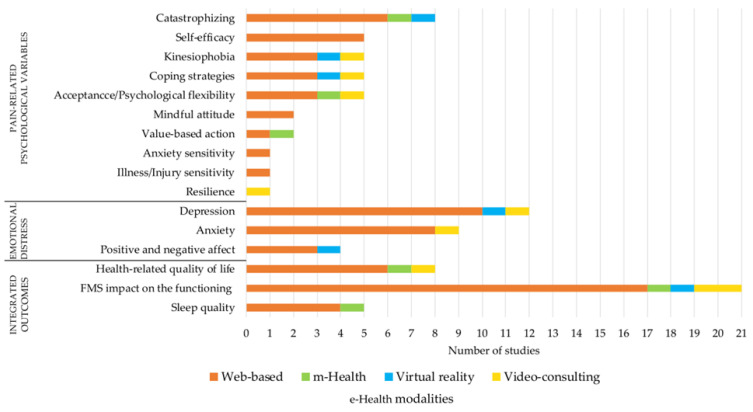
Overview of variables assessed in the included studies by each e-Health modality.

**Table 1 healthcare-11-01845-t001:** Eligibility criteria according to the PICOs model.

PICOs	Inclusion Criteria	Exclusion Criteria
Population (P)	Adults (i.e., mean age of the sample ≥ 18 years of age) Diagnosis of FMS (i.e., sample of patients with FMS or sample with chronic pain in which the percentage of patients with FMS is specified and higher than 50% of the total sample *)	Sample with children or adolescents only (i.e., ≤18 years of age) Sample of patients without a diagnosis of FMS or sample in which the percentage of patients with FMS is unspecified or lower than 50% of the total sample
Intervention (I)	Psychological or multicomponent interventions implemented (even partially) through an e-Health modality	No psychological or multicomponent interventions; no e-Health modalities
Comparison (C)	When available, no limitations were set on the comparison conditions	None
Outcomes (O)	At least one psychosocial outcome or pain-related psychological variable or health-related integrated variables with at least a psychosocial/emotional component	Only physical outcome or strictly pain-related outcome without any psychosocial or emotional component
Study design (S)	Qualitative and quantitative research without any restrictions	None

* In this case, data relating to only patients with FMS are reported, or if this is not possible, the data of total sample is reported and the percentage of patients with FMS out of the total is indicated.

**Table 2 healthcare-11-01845-t002:** Assessment of studies’ quality based on the QATSDD method.

Study	Explicit Theoretical Framework	Statement of Aims/Objectives in Main Body of Report	Clear Description of Research Setting	Evidence of Sample Size Considered in Terms of Analysis	Representative Sample of Target Group of a Reasonable Size	Description of Procedure for Data Collection	Rationale for Choice of Data-Collection Tool(s)	Detailed Recruitment Data	Statistical Assessment of Reliability and Validity of Measurement Tool(s) *	Fit Between Stated Research Question and Content of Data-Collection Tool °	Fit Between Stated Research Question and Method of Data Collection *	Fit Between Stated Research Question and Method of Analysis	Assessment of Reliability of Analytical Process °	Good Justification for Analytical Method Selected	Evidence of User Involvement in Design	Strengths and Limitations Critically Discussed	QATSDD Total Score
Lorig et al. [28]	3	3	2	0	2	3	3	3	3		3	3		3	0	3	34
Williams et al. [29]	3	3	3	3	2	3	3	1	0		3	3		3	0	3	33
Morris et al. [30]	3	3	3	0	2	3	2	3	3		3	3		3	0	0	31
Kristjánsdóttir et al. [31]	3	3	3	3	2	3	1	3	3		3	3		3	1	3	37
Botella et al. [32]	3	3	3	0	1	1	1	3	2		2	2		2	0	2	25
Camerini et al. [33]	1	3	2	0	3	2	2	2	1		2	3		3	3	3	30
Davis & Zautra [34]	3	3	2	3	2	3	1	2	3		2	3		3	0	3	33
Menga et al. [35]	3	3	3	0	2	3	1	3	0		1	3		3	0	2	27
Ljótsson et al. [36]	3	3	2	3	2	3	1	3	1		3	3		3	0	3	33
Vallejo et al. [37]	3	3	3	3	2	2	2	3	3		3	3		3	0	2	35
Sparks et al. [38]	3	3	3	1	3	3	2	3	3		2	3		3	0	2	34
Friesen et al. [39]	3	3	3	3	2	3	1	3	3		3	3		3	0	3	36
de la Vega et al. [40]	3	3	2	3	2	3	1	2		3		2	3	2	3	2	34
Molinari et al. [41]	3	3	3	3	2	3	1	3	3		3	3		3	0	3	36
Yuan & Marques [42]	1	3	2	0	1	2	1	1		2		2	0	2	3	0	20
Simister et al. [43]	3	3	2	3	2	3	1	3	3		3	3		3	1	3	36
Hedman-Lagerlöf et al. [44]	3	3	3	3	2	3	3	3	3		3	3		3	1	3	39
Friedberg et al. [45]	2	3	2	3	2	2	2	3	3		3	3		2	0	2	32
Collinge et al. [46]	0	3	2	2	2	3	2	3	3		2	3		3	0	2	30
Carleton et al. [47]	1	2	2	3	2	3	1	2	3		3	3		2	0	3	30
Climent-Sanz et al. [48]	3	3	3	3	3	2	3		3	3	3	3	3	3	0	3	35 ^1^
Serrat et al. [49]	3	3	3	1	2	2	3	3	3		3	3		3	0	3	35
García-Perea et al. [50]	1	3	3	3	2	3	1	2	1		3	3		3	0	2	30
de la Coba et al. [51]	3	3	3	1	1	3	3	2	3		3	3		3	0	3	34
Ong et al. [52]	3	3	3	3	2	3	2	3	1		3	3		3	0	2	34
Paolucci et al. [53]	3	3	3	0	1	3	1	2	2		3	3		3	0	3	30
Mean value	2.5	3.0	2.6	1.9	2.0	2.7	1.7	2.6	2.3	2.7	2.7	2.9	2.0	2.8	0.5	2.4	32.4

Score: 0 = not at all; 1 = very slightly; 2 = moderately; 3 = complete. * Applies only to quantitative studies; ° applies only to qualitative studies. ^1^ The mean score was calculated by subtracting from the total mean score the mean score of columns “Statistical assessment of reliability and validity of measurement tool(s)” and “Fit between stated research question and content of data collection tool, e.g., interview schedule” plus the mean score of columns “Fit between stated research question and method of analysis” and “Assessment of reliability of analytical process” since this study included both qualitative and quantitative aspects.

**Table 3 healthcare-11-01845-t003:** e-Health modalities, main psychological approaches, and strategies at the basis of the interventions.

e-Health Modality	Reference	Type of Intervention	Intervention Details
WEB-BASED PROGRAM			
(a) Only web-based	Williams et al. [29]	CBT-based self-management program for symptom adaptive lifestyle management (web-enhanced behavioral self-management—WEB SM)	Contents and structure: Thirteen modules contained in a website and segregated into three broad segments: educational lectures providing background knowledge about FMS; education, behavioral, and cognitive skills designed to help with symptom management; and behavioral and cognitive skills designed to facilitate adaptive lifestyle changes for managing FMS. Duration: 6 months
	Davis & Zautra [34]	Mindfulness-based socioemotional regulation intervention for coping with pain and stress, positive and negative affect, and positive engagement in social relations	Contents and structure: Twelve modules delivered via Adobe Presenter and centered on the following topics: welcome and introduction to mindfulness; positive and negative emotions; mindfulness of emotions; awareness of emotions and pain; acceptance of emotions; mindful living with pain; pacing yourself mindfully; emotions and thoughts; thoughts and beliefs; savoring the positive; building mindful relationships; and bringing it all together. Duration: 6 weeks
	Menga et al. [35]	CBT and interpersonal-therapy-based program to prevent and manage depression and anxiety symptoms (MoodGYM)	Contents and structure: Five modules delivered via a website based on cognitive reconstructing, relaxation, pleasant events, assertiveness training, and problem-solving. Duration: 6 weeks
	Sparks et al. [38]	Psychoeducation intervention to improve knowledge about FMS, skills for symptom management, and adopt healthier lifestyles (FibroGuide)	Contents and structure: Ten modules delivered via DVD format or USB flash drives (since the program was not available via the internet during study implementation) and including the following components: understanding FMS; communicating with family and healthcare providers; being active; improving sleep; relaxing; coping with “fibro fog” (cognitive difficulties); setting goals; pacing self; thinking differentially; and making time for self. Duration: 12 weeks
	Friedberg et al. [45]	Self-management intervention based on bilateral stimulation and desensitization (BSD) targeting pain and fatigue reduction	Contents and structure: BSD technique involving gentle bilateral stimulation using soft audio sounds (via earphones) or gentle finger tapping on the upper legs while the patient focuses on their most salient pain. After an initial training session, BSD was delivered for pain and stress reduction through a BSD demonstration video, duplicate written instructions, and an mp3 file of the audio BSD technique. Online diary tracked pain, fatigue, and intervention use. Duration: 90 days
	Climent-Sanz et al. [48]	Therapeutic patient education intervention for pain and sleep	Contents and structure: The intervention was delivered via a website according to the following plan: completing a socio-demographic data sheet and questionnaires; watching a video about the objective, characteristics, and functions of the web platform; watching a video where FMS and its associated symptoms were validated; focus on cognitive factors (week 1 and 2); behavioral factors (week 3 and 4); different activities to challenge participants to implement daily management strategies for pain and poor sleep quality based on what they learned during the previous 4 weeks; and access to a personal diary (week 5 and 6). Duration: 6 weeks
(b) Web + automated feedbacks	Molinari et al. [41]	Positive future-thinking intervention for promoting positive affect and functioning (best possible self intervention—BPS)	Contents and structure: In a face-to-face session, patients were asked to imagine and write down their future best possible self through a computer application and then to visualize what they had just written. Patients were instructed to continue visualizing their BPS at home, accessing a dedicated web platform. During the intervention, participants received two automatic weekly SMS reminders to practice their exercises and reinforcements. Duration: 4 weeks
	Collinge et al. [46]	A self-management program targeting the functional impact of FMS	Contents and structure: Participants accessed a website using an online health diary program (“SMARTLog”) to report symptom ratings, behaviors, and management strategies. The automated feedback program provided individualized recommendations based on a single-subject analysis of the accumulated data over time (“SMARTProfile”). Duration: 11-month
(c) Web + interaction with live professionals’ feedbacks	Camerini et al. [33]	Self-management education intervention to increase knowledge and empowerment (ONESELF)	Contents and structure: The intervention was delivered via a website, which included the following functionalities: a virtual library providing users with relevant information on the disease; a first aid and a frequently asked questions section (FAQ), providing brief and practical information on the syndrome management; and a virtual gymnasium providing patients with tailored multimedia contents on several physical exercises. The website also enabled asynchronous and synchronous interactions with health professionals and laypeople to provide social support. Duration: 167 days
	Ljótsson et al. [36]	Acceptance and values-based exposure treatment to improve quality of life, psychological flexibility, and health-related costs	Contents and structure: The intervention material was presented on printer-friendly web pages and divided into five successive steps: introduction with information about pain and FMS; explanation of the learning of symptom-related fear of how FM-related avoidance behaviors maintain fear and ultimately lead to disability; promotion of a mindful and accepting stance towards negative thoughts and experiences; continued values-based behavioral change through exposure; and relapse management. Participants were encouraged to send at least a weekly message about their work with the intervention to their therapist, who could, in turn, contact them through text messages. Duration: 10 weeks
	Hedman-Lagerlöf et al. [44]	Exposure therapy to break the vicious cycle of preoccupation with symptoms, avoidance, and increased pain	Contents and structure: The intervention was delivered by a web platform and divided into eight modules, to which the participant gained gradual access by completing homework assignments: role of avoidance behaviors; psychoeducation about exposure; identification of personal avoidance behaviors; planning phase; and personalized exposure exercises. Treatment progress was closely monitored by a therapist, with whom participants had regular (about one-to-three times/week) contact through asynchronous text messages on the platform. Duration: 10 weeks
	Serrat et al. [49]	Multicomponent intervention (therapeutic exercise, pain neuroscience education, CBT, and mindfulness training) for reducing functional impairment and improving several psychological and physical variables (FIBROWALK)	Contents and structure: Participants were emailed a link to a 60 min video on a private YouTube channel once a week. Each video provided detailed guidelines explaining how to perform different home-based aerobic exercises, education in the neuroscience of pain, CBT to decrease anxiety and depressive symptoms, pain catastrophizing, and changing inadequate emotional regulation strategies. Participants could contact the therapist (via email) at any time. The therapist could also contact them in case of any issue related to the program or the study. Duration: 12 weeks
	Simister et al. [43]	ACT-based intervention for reducing the FMS impact on daily functioning and improving pain, mood, and physical function	Contents and structure: The intervention was delivered via a website on seven units: introduction; acceptance; values; medications, sleep, “fibro fog”, exercise, and effective communication; cognitive defusion (or you are not your thoughts!); mindfulness and self-as-context; and are you willing? Written unit materials were provided in PDF format and enhanced through mp3 audio recordings, videos, and experiential homework exercises. The treatment team provided participants with weekly email reminders to complete the program and a reminder to contact a team member if they had any concerns. Duration: 8 weeks
	García-Perea et al. [50]	Self-management education intervention to improve quality of life (Red Sinapsis)	Contents and structure: A web platform provided patients with information on their illnesses, access to their clinical history, and a messaging system for communicating with the medical or nursing staff at any time. Duration: 12 months
	Ong et al. [52]	Positive affect (PA) skills-building intervention program to keep stress and pain under control (LARKSPUR)	Contents and structure: The intervention was delivered via a website and included eight PA skills over five learning modules: noticing positive events; savoring positive events; identifying personal strengths; behavioral activation; mindfulness and positive reappraisal; gratitude; and acts of kindness. The intervention was supplemented by support from research staff via telephone and email. Duration: 5 weeks
(d) Web + automated and live feedbacks	Lorig et al. [28]	The self-management program focused on reducing pain and improving functioning	Contents and structure: The program was delivered via a website targeting the following components: self-management principles; goal setting/action plans; pain management; problem-solving steps; fitness/exercise; feedback/problem-solving; difficult emotions; healthy eating; osteoporosis; fatigue and energy conservation; medication; depression; work with your health care professional; evaluating treatment plans; and sleep. Participants were asked to log on at least three times for 1-2 h and participate in the weekly activities. Any problem they wished to discuss could be posted on the bulletin board and responded to by other group members and the moderators. The program also used email reminders to encourage nonparticipants to participate. Duration: 6 weeks
	Vallejo et al. [37]	CBT intervention for reducing the FMS impact on daily functioning and improving different pain-related psychological variables	Contents and structure: The intervention was delivered via a web application and included ten modules: psychoeducation; progressive relaxation training; emotional training; increasing and adjusting daily activities; techniques for insomnia and sexual dysfunctions; problem-solving; cognitive restructuring; attentional control and illness behaviors; intellectual problems; and revision and relapse intervention. Participants could send individual messages to the therapist. The program had several points to facilitate interaction with the professional (e.g., feedback message to the participant to reinforce the weekly schedule). Duration: 10 weeks
	Friesen et al. [39]	CBT based self-management program addressing pain, disability, and emotional wellbeing (The Pain Course)	Contents and structure: The intervention was delivered via a website and included five weekly lessons, homework, and additional resources. The following components were included: prevalence of chronic pain and symptoms of depression and anxiety; information about pain perception; cognitive behavioral model; principles of cognitive therapy; strategies for monitoring and challenging thoughts; physical symptoms of anxiety and depression, chronic pain; controlled breathing; pleasant activity scheduling; pacing; graded exposure; relapse prevention; and goal setting. Clinical contact with participants occurred weekly via secure messaging and telephone to provide general support and encouragement. They also received standardized automated emails each week. Duration: 8 weeks
(e) Dedicated computer software	Carleton et al. [47]	Attention bias modification (ABM) computer program for reducing patients’ hypervigilance for pain-related cues	Contents and structure: Participants were given attention tasks using word stimuli established as relevant to pain-specific attentional biases and matched to neutral words of comparable length. At each treatment session, participants were required to rate their emotional intensity associated with each of the 48 pain-specific threat words from “not at all bothersome” to “very bothersome”. The computer then used the 20 words rated as most negative by each participant as the threat words for that session, which should have facilitated personal relevance. Duration: 4 weeks
MOBILE PHONE APPLICATION (m-Health)	Kristjánsdóttir et al. [31]	ACT intervention to reduce catastrophizing and improve function	Contents and structure: The intervention was delivered via mobile software after participants completed an inpatient multidimensional rehabilitation program for chronic pain and included the following components face-to-face sessions; web-based diaries; written situational feedback from a therapist; and audio files with guided mindfulness exercises. Duration: 4 weeks
	Yuan & Marques [42]	Multicomponent intervention (psychoeducation, CBT strategies, and physical activity) to improve HrQoL, symptoms, and self-care agency (ProFibro)	Contents and structure: The intervention was delivered via a mobile phone application. It included the following components: patient education through animation, self-monitoring, sleep strategies, scheduling, graded exercise program, gratitude practice, family adjustments, and hints. Duration: 1 week
	de la Vega et al. [40]	CBT intervention to improve the quality of life (Fibroline)	Contents and structure: The intervention was delivered via a mobile phone application. It included different modules targeting the following components: life values and goal setting, sleep quality, anxiety management, pain education and coping, medication use, physical conditions, mood regulation, thoughts management, and relapse prevention. Four types of tasks were activated when treatment modules were accessed: resources; assessments; notes; and reminders. Duration: 9 weeks
VIRTUAL REALITY (VR)	Morris et al. [30]	Exposure therapy program as a treatment for exercise-related pain catastrophizing	Contents and structure: Visual exposure to exercise activities delivered via a VR head-mount display (HMD) as part of an in-person intervention. Duration: 16 weeks
	Botella et al. [32]	CBT intervention to improve pain- and mood-related variables	Contents and structure: A VR system was used as an adjunct to face-to-face CBT for delivering relaxation and mindfulness. In more detail, while participants were immersed in the VR, the system provided instructions on observing the different elements offered by the scenarios, remaining focused on the present moment and participating in the experience without making any judgments. Duration: 6 weeks
VIDEO CONSULTING	de la Coba et al. [51]	ACT intervention to enhance the patient’s openness to experiencing pain and associated emotional discomfort	Contents and structure: Online groups of Google Meet video-meeting sessions (105 min. duration each), where each consisted of an initial presentation (or participatory summary of the previous session), a review of experiences after practicing the proposed activities at home, a presentation of metaphors and practice of experiential exercises, scheduling of practice activities at home, reflections, and resolution of doubts and queries. Duration: 5 weeks
	Paolucci et al. [53]	Mind–body intervention to improve pain, function, and different pain-related psychological variables	Contents and structure: Video-meeting sessions (60 min. duration each) using an online communication platform (i.e., Zoom) based on the following principles: anchoring to a positive emotion through the choice of a color, “here and now”: listen and perceive your “own” body in motion, conscious breathing, “close your eyes”: improve interoceptive awareness during physical exercises, and relaxation: breathe, moving slowly and without pain. Duration: 8 weeks

ACT: acceptance and commitment therapy; CBT: cognitive behavioral therapy; FMS: fibromyalgia syndrome; HrQoL: health-related quality of life; VR: virtual reality.

## Data Availability

Not applicable.

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
