# Peer review of "e-Health Interventions Targeting Pain-Related Psychological Variables in Fibromyalgia: A Systematic Review"

_healthcare, 2023, doi:10.3390/healthcare11131845_

Round 1

Reviewer 1 Report

Even though the study was interesting, I noticed serious flaws in the study selection and result writing. So I didn’t read the discussion section.

Some specific comments are below

·         If already a study on the same topic is available which was conducted in 2019 what is the need for this study?

·         The search strategy, research question, and eligibility criteria are not based on PICO.

·         Why was the paper with low QATSSD included in the review?

·         Line 167-170 – six articles were excluded as the same researchers used them in a previous study. This may cause serious bias in the results.

·         The population selected in the study is heterogenous, and even all the studies were not of participants with FMS.

·         Why is there no age restriction in the population 

Author Response

Dear Reviewer, many thanks for your comments, which allow us to improve the quality of our manuscript. Below, we have reported our answers for each comment. All the changes that we made to the manuscript are visible in track-change modality.

  1. If already a study on the same topic is available which was conducted in 2019 what is the need for this study?

RESPONSE: We thank the Reviewer for this question. As already specified in the introduction of the manuscript, one review on the same topic exists, but it included only papers with an RCT design. Moreover, results focused only on Internet-based cognitive behavioral therapies (ICBTs). Considering the increased interest for the e-Health solutions in the current years (also even during and after the pandemic) with our paper we aimed at updating and enlarging the scope of this previous systematic review, including all the potential e-Health psychological and multicomponent interventions for patients with FMS, with no restrictions on the study design.

  1. The search strategy, research question, and eligibility criteria are not based on PICO.

RESPONSE: Following the Reviewer’s suggestion, the search and study selection strategies according to PICO have now been added in the paragraph “2.2 Inclusion and Exclusion Criteria” of the Method section.

  1. Why was the paper with low QATSSD included in the review?

RESPONSE: We have not ruled out the paper with low QATSSD because our aim was to give a comprehensive and general overview of the current state-of-art in terms of quality in this field and help the reader to understand the level of quality of the current literature.

  1. Line 167-170 – six articles were excluded as the same researchers used them in a previous study. This may cause serious bias in the results.

RESPONSE: We have excluded 9 articles because they analyzed e-Health interventions already included and described in previous articles (which were included in the review) by the same research group (i.e., further analysis of the same intervention). In these cases, the first paper presenting the intervention has been included in the review, while the others have been excluded from the description of results in terms of numbers (to avoid bias) but significant data included in those secondary papers were inserted in Table 1 and, if relevant, discussed in the paper.

  1. The population selected in the study is heterogenous, and even all the studies were not of participants with FMS.

RESPONSE: We have included patients with FMS without any restrictions. Studies regarding patients with chronic pain/disease in which the percentage of patients with FMS was unspecified or lower than 50% were excluded. If a study is included, data relating to only patients with FMS are reported, or, only in the case this is not possible, data from the total sample are reported and the percentage of patients with FMS out of the total is indicated. This explanation is reported even in the manuscript.

  1. Why is there no age restriction in the population.

RESPONSE: As stated in the paragraph “2.2 Inclusion and Exclusion Criteria” of the Method section, we have included adult participants (i.e., the mean age of the sample higher than 18 years or older) diagnosed with FMS.

Reviewer 2 Report

I found the manuscript fascinating to read. It was well-written and well-thought-out. The methodology was sound, and the procedures were well-defined. The findings were clearly presented and critically analysed. This article should without a doubt be considered for publishing in this prestigious journal. Congratulations to all of the authors.

Author Response

Thank you for you comments, they are much appreciated.

Reviewer 3 Report

This paper presents a systematic literature review on internet interventions for fibromyalgia.

Suggestions and questions (answers may be used to improve the manuscript):

1. Check mdpi.com/journal/healthcare/instructions - "The abstract should be a total of about 200 words maximum."

2. This review must be updated; '16 August 2022' is outdated.

3. Table 1 should be presented in the results section, because it presents results, right?

4. What are the research questions of this systematic review? They should be declared.

5. Consider "Efficacy of the e-Health interventions"; what is 'efficacy' in this context?

6. Results section only explores the use of tables. Charts or figures are not explored. Also, the results section is too verbose and, sometimes, it gets boring to read.

7. Discussion could provide trends and open issues, which are relevant to readers.

8. Title could be 'straight to the point'. For example, "e-Health Interventions for Fibromyalgia: a Systematic Literature Review"

9.  Figure 1 is NOT the PRISMA Flow Diagram; it is BASED ON PRISMA Flow Diagram.

1. Avoid short paragraphs or paragraphs with a single sentence (e.g., lines 221-223)

2. line 158: "Doubts were discussed, and, where necessary,..." -> WHEN necessary

Author Response

Dear Reviewer, many thanks for your comments and suggestions, which allow us to improve the quality of our manuscript. In the file, below, we have reported our answers for each comment. All the changes that we made to the manuscript are visible in track-change modality.

  1. Check mdpi.com/journal/healthcare/instructions - "The abstract should be a total of about 200 words maximum."

RESPONSE: We thank the Reviewer for this suggestion. We have reorganized the abstract in order to limit the length to 200 words.

  1. This review must be updated; '16 August 2022' is outdated.

RESPONSE: Following the Reviewer’s suggestion, we have updated the review by using the same research strategies, with a new search for papers in the time span between August 17 2022 and May 15 2023, and we have modified the manuscript’s contents accordingly. The updating literature search yielded 103 records in total, with two duplicates that were removed. One-hundred and one (101) records were analyzed by title and abstract, and 96 were excluded according to inclusion and exclusion criteria. Finally, five records were selected for the full-text analysis, of which four were excluded for various reasons (i.e., no psychosocial outcomes or pain-related psychological variables; secondary analysis of the main included papers). In the end, only a new article (Paolucci et al., 2022) met the inclusion criteria and thus was included in addition to the previously included 25 papers (Flow-chart - Figure 1 has been updated).

  1. Table 1 should be presented in the results section, because it presents results, right?

RESPONSE: We have moved Table 1 to the Results section and adapted the references accordingly.

  1. What are the research questions of this systematic review? They should be declared.

RESPONSE: According to the aims of the manuscript, we have specified more clearly the research questions guiding our systematic review at the beginning of the Method section as follows:

Three main research questions guided the current review:

  • What e-Heath tools are under investigation to deliver psychological and/or multicomponent interventions targeted psychosocial outcomes and/or pain-related psychological variables in patients with FMS?
  • What are the main characteristics of those e-Health interventions in terms of underlying psychological approaches, structure and addressed outcomes?
  • What is the impact of such e-Health tools in terms of signals of efficacy, feasibility, and acceptability?
  1. Consider "Efficacy of the e-Health interventions"; what is 'efficacy' in this context?

RESPONSE: According to Reviewer’s suggestion, at the end of the paragraph “2.4 Data Extraction and Synthesis” we have added the following specification to clarify what we mean for “efficacy” in this context and adapt the text accordingly: “Moreover, signals of efficacy, feasibility and acceptability have been summarized, considering the available results (even in secondary papers). It should be specified that this article aims at giving a broad overview of the current state of knowledge on signals of efficacy in that specific field. For signal of efficacy we consider results on the impact of the interventions on different outcome measures, without claiming to provide a definitive or peremptory conclusion in that regard. As for feasibility and acceptability, any information reported on those aspects have been synthetized”.

  1. Results section only explores the use of tables. Charts or figures are not explored. Also, the results section is too verbose and, sometimes, it gets boring to read.

RESPONSE: We have restructured the Results section according to the Reviewer’s suggestions. More specifically, we added a figure to sum up the types of e-Health modalities (Figure 2) and a bar chart to give an overview of the variables targeted in the included studies by each e-Health modality (Figure 3). Moreover, we synthetized the paragraph “3.5.3 Psychosocial outcomes and pain-related psychological variables targeted by the e-Health interventions”, moving the descriptions of the outcome measures (i.e., questionnaires) used in the studies to Table S2 reported in Appendix 2.

  1. Discussion could provide trends and open issues, which are relevant to readers.

RESPONSE: According to Reviewer’s suggestion, at the end of the “Discussion” section, we have summarized and reported the main open issues into this field (from line 632 onward).

  1. Title could be 'straight to the point'. For example, "e-Health Interventions for Fibromyalgia: a Systematic Literature Review".

RESPONSE: Following the Reviewer’s suggestion, we have modified the title as follows: “e-Health Interventions Targeting Pain-related Psychological Variables in Fibromyalgia: a Systematic Review”.

  1. Figure 1 is NOT the PRISMA Flow Diagram; it is BASED ON PRISMA Flow Diagram.

RESPONSE: We have modified the description of Figure 1 according to the Reviewer’s suggestion as follows: “Figure 1. Process of Study Selection (based on PRISMA Flow Diagram).

Comments on the Quality of English Language

  1. Avoid short paragraphs or paragraphs with a single sentence (e.g., lines 221-223)

RESPONSE: We have rearranged the text to avoid short paragraphs or paragraphs with a single sentence (e.g., lines 238-240; 256-257; 272-273; 331-333; 334-337; 346-349,…)

  1. line 158: "Doubts were discussed, and, where necessary,..." -> WHEN necessary

RESPONSE: According to the Reviewer suggestion, we have restructured the sentence as follows: “Doubts were discussed, and any disagreement about study eligibility was resolved by a third reviewer”.

Round 2

Reviewer 1 Report

I noticed several major issues with this manuscript. I did not go through the discussion section. Once these issues are resolved, only then will I read the discussion.

As there is already one recent systematic review available  with high-quality evidence (RCT), the need for this study must be clearly mentioned in the introduction.

Search criteria are not based on PICO

The researcher excludes six articles without specific reason – this will influence the study results. 

Author Response

Dear Reviewer, many thanks for your comments, we are sorry if our previous answers on those topics were not sufficiently complete, we now try to improve these changes. All the changes that we made to the manuscript are visible in track-change modality.

1. As there is already one recent systematic review available with high-quality evidence (RCT), the need for this study must be clearly mentioned in the introduction.

RESPONSE: We had already specified in the introduction of the first version of our manuscript that one review on the same topic exists, also reporting the reasons for conducting this new review. Specifically, our review aims to 1) update and 2) enlarge the scope of the review by Bernardy et al. 2019. As for the former aim, the previous review has searched papers until January 2018. Considering the dramatically increased interest in e-Health approaches in the healthcare context in the last 5 years (mainly, but not only, due to the pandemic), our systematic review seems extremely relevant (of note, please consider that in revising the manuscript we have updated the literature search until May 2023). As for the second aim, the review by Bernardy and colleagues included only papers with an RCT design, focusing on Internet-based solutions and cognitive behavioral therapies (ICBTs). We think that the more inclusive approach we adopted can provide a more detailed overview of the state of the art in the field of psychosocial interventions in FMS, using e-health solutions. Compared to the first version of our manuscript, now we have further developed these contents in the introduction section (see lines 80-86). We hope that such changes make the need for this study clearer.

2. Search criteria are not based on PICO.

RESPONSE: Thank you for this comment. In this revised version we have further developed the PICO section (see Paragraph 2.2), adding some details to the text, and creating a table that explicitly refers to the PICO approach, describing the inclusion/exclusion criteria. See table 1. We hope that these further details are in line with the reviewer's request.

3. The researcher excludes six articles without specific reason – this will influence the study results. 

RESPONSE: Thank you for this comment which suggests we need to be clearer in the description of our methodology. Being the focus of our review the identification and description of the existing e-health interventions, we have preferred to include the papers only the first time that the intervention was studied/described (i.e., “primary paper”). Therefore, we have consistently excluded 9 articles (please note that with the update of the review timeline up to May 2023, further 3 papers have been added as “secondary analysis papers”) from the count of the included papers (e.g., in the final number of the included papers in the flowchart, in the results of sections 3.2, 3.3, 3.4, in the numbers reported in Figures 2 and 3, and whenever we needed to quantify the included papers). We think that such approach prevents us to double-count an e-health intervention, finally avoiding biases. We preferred to consider the first paper as in that case there was the main description/information of the intervention.

As an example, the reader should be aware that 3 mobile-phone applications have been applied so far. In case we would have included in our flow chart the secondary analysis paper by Mirò et al. 2022 (focusing on the same mobile-phone app of the primary paper by de la Vega et al. 2018) without distinguishing between primary and secondary analysis papers, the reader would have a false idea about the number (and therefore, the state of the art) of mobile-phone applications nowadays existing for FMS.

Please also consider that, to avoid missing relevant data potentially described in the so-called ‘secondary analysis papers’, we inserted them in Table S2 and in the results/discussion. In this way, the reader has the possibility to distinguish between primary and secondary analysis papers and to be updated on further references/data related to each intervention after its first publication.

We have further detailed this issue in the revised version of the manuscript (paragraphs 2.2 and 3.1).

Reviewer 3 Report

The authors addressed most of my comments. Only one remains:

My comment: What are the research questions of this systematic review? They should be declared.

RESPONSE: According to the aims of the manuscript, we have specified more clearly the research questions guiding our systematic review at the beginning of the Method section as follows:

“Three main research questions guided the current review: ..."

Reviewer's reply: Why are they the main ones? Are there others? If so, they must be declared.

Author Response

Thanks for this comment, we have deleted the misleading word “main”, as those written are all the research questions.